# Importance of Potential New Biomarkers in Patient with Serouse Ovarian Cancer

**DOI:** 10.3390/diagnostics11061026

**Published:** 2021-06-03

**Authors:** Aneta Cymbaluk-Płoska, Karolina Chudecka, Anita Chudecka-Głaz, Katarzyna Piotrowska, Sebastian Kwiatkowski, Maciej Tarnowski

**Affiliations:** 1Department and Clinic of Gynaecological Surgery and Gynaecological Oncology of Adults and Adolescents, Pomeranian Medical University, ul. Powstańców Wlkp. 72, 70-111 Szczecin, Poland; karolina.chudecka@gmail.com (K.C.); anitagl@poczta.onet.pl (A.C.-G.); 2Department of Physiology, Pomeranian Medical University, Powstańców Wlkp. 72, 70-111 Szczecin, Poland; piot.kata@gmail.com (K.P.); maciejt@pum.edu.pl (M.T.); 3Department of Obstetrics and Gynaecology, Pomeranian Medical University, ul. Powstańców Wlkp. 72, 70-111 Szczecin, Poland; kwiatkowskiseba@gmail.com

**Keywords:** ovarian cancer, YAP, TEAD4, SMAD2, SMAD3, H2A.X, ALD1A1, CD71, TKT, TKTL1

## Abstract

Ovarian cancer remains the gynecological cancer with the highest mortality rate. In our study, we compare a number of proteins from different effector pathways to assess their usefulness in the diagnosis of ovarian cancer. The tissue expression of the tested proteins was assessed by two methods: qRT-PCR and an immunohistochemical analysis. A significantly higher level of mRNA expression was found in the ovarian cancer group for *YAP* and *TEAD4* (*p* = 0.004 and *p* = 0.003, respectively). There was no statistical significance in the expression of mRNA for *SMAD3*, and there was borderline statistical significance for *SMAD2* between the groups of ovarian cancer patients and other subgroups of patients with simple cysts and healthy ovarian tissue (*p* = 0.726 and *p* = 0.046, respectively). Significantly higher levels of transferrin receptor (*CD71*), *H2A.X*, and *ADH1A* gene expression were found in the ovarian cancer group compared to the control group for *YAP*, and TEAD4 showed strong nuclear and cytoplasmic staining in ovarian carcinoma and weak staining in non-carcinoma ovarian samples, ADH1A1 showed strong staining in the cytoplasm of carcinoma sections and a weak positive reaction in the non-carcinoma section, H2A.X showed strong positive nuclear staining in carcinoma sections and moderate positive staining in non-carcinoma samples, and CD71 showed moderate positive staining in carcinoma and non-carcinoma samples. YAP, TEAD4, and ADH1A proteins appear to be promising biomarkers in the diagnosis of ovarian cancer.

## 1. Introduction

Epithelial ovarian cancers (EOCs) constitute the majority of malignant ovarian cancers in adult women. The diagnosis of ovarian cancer has not improved over the years. Only a small percentage of patients visit a gynecologist at an early stage of the disease due to the lack of characteristic symptoms. The prognosis of patients with advanced disease is very serious.

Conventional therapies usually include surgery and chemotherapy (platinum and taxol-based, as well as radiotherapy). The literature includes many proteins with their expression assessed in ovarian cancer tissue and compared with healthy ovarian tissue. There are ongoing searches for proteins that could serve as diagnostic, prognostic, or predictive factors in the fight against this cancer. The expression of proteins involved in apoptosis, proliferation, DNA repair, and immune processes are of specific interest in ovarian cancer research.

Many current studies concern the role of SMAD-group proteins. SMADs are proteins that are signal transmitters and transcription modulators that mediate numerous signaling pathways. In ovarian cancer patients, mutations in the *SMAD* family of proteins are not common, but they have been found in cellular lines and primary cultures [1]. Inactivation of those proteins is associated with a more aggressive course of this type of tumor [2]. In previous studies, we found that the expression of the *SMAD4* gene significantly differed in healthy patients compared to those with ovarian cancer [3]. It was also an independent predictive factor. Higher expression of the SMAD4 protein was associated with a shorter overall survival (OS) of patients with ovarian cancer.

The Hippo pathway is a recently discovered signal transduction pathway. YAP promotes cell proliferation, inhibits cell apoptosis, and also promotes the endodermal-mesenchymal transition (EMT) of cells [4]. YAP plays a key role in the development and progression of many types of cancer, including ovarian cancer. However, the impact of YAP on the in vivo development of ovarian cancer and its effects remains uncertain. Xiu et al. were the first to show that high levels of *YAP* expression were positively correlated with the expression of the *TEAD4* gene [5]. 

*TEAD* binds to gene promoter sites, which are responsible for inhibiting apoptosis or promoting cell proliferation. Research confirmed that the YAP molecule is required for the full transcriptional activity of this protein, with the YAP molecule always co-precipitating with the TEAD factor [6,7]. Data in the literature confirmed the role of Hippo pathway components in the tumorigenesis of various types of tissues, also showing the complicated relations between Hippo and other functionally similar biochemical pathways [8].

Double-strand breakage (DSB) is one of the first procedures that initiates tumor formation under the influence of both endogenous and exogenous factors [9]. Several years ago, it was discovered that phosphorylation of the histone protein H2A.X in the serine 139 position (after phosphorylation, marked as *γ-H2A.X*) during DSB formations is important in the formation of a repair complex of double-strand DNA breaks. The increased expression of *γ-H2A.X* could potentially serve as a biomarker for the transformation of normal tissue into the pre-neoplastic condition and, consequently, neoplastic tissues [10].

ALDH1 is one of the isoenzymes of aldehyde dehydrogenase responsible for the oxidation of acetaldehyde. Its toxic and potentially carcinogenic effects have been described for many years. It has been suggested that neoplastic cells have a high ethanol oxidation capacity but a lower acetaldehyde removal capacity compared to normal tissues [11]. This further intensifies the proliferation process and disturbs the metabolism of some biologically important factors, e.g., retinoic acid. It has been suggested that the determination of changes in serum and tissue levels of ALDH1 isoenzymes could be useful in the diagnosis of some cancers [12].

The transferrin receptor (CD71) is present at a low level in almost all tissues in our body. CD71 is a type II receptor that is located on the outer cell membrane. It transports iron in the form of clarine or dynamin-dependent endosomes and then returns to the surface of the cell [13]. Despite the fact that CD71 is almost ubiquitous in our body, its expression is much stronger in cancerous tissues. In the case of some cancers, it is closely related to tumor advancement or differentiation [14].

Transketolase (TKT) is a key enzyme in the anaerobic pentose phosphate pathway. There are three isoenzymes: TKT, TKTL1, and TKTL2. All of the enzymes belonging to the transketolase family are found in normal tissues as well as in cancerous tissues. Increased glycolysis and the pentose phosphate pathway are characteristic for cancer cells to maintain these cells in homeostasis during oxidative stress and to quickly provide energy for fast division and, hence, for proliferation and metastasis [15,16].

### Aim of the Study

We evaluated the differences in the expression of the *YAP, TEAD4, SMAD2, SMAD3, H2A.X, ALD1A1, CD71, TKT*, and *TKTL1* in ovarian cancers and benign cysts. We evaluated the usefulness of *YAP, TEAD4, SMAD2, SMAD3, H2A.X, ALD1A1, CD71, TKT* and *TKTL1* as diagnostic or prognostic markers in patients with ovarian cancer.

## 2. Materials and Methods

### 2.1. Tissue Specimens

The specimens were obtained from 79 women who were subject to surgeries in the years 2015–2016 in the Department of Surgical and Oncological Gynecology of the Pomeranian Medical University. The types of surgeries performed providing us with tissues included bilateral adnexectomy or hysterectomy with bilateral adnexectomy. Tissues stored in −70 °C and histopathological specimens that were prepared by fixation in 10% formalin were used in the study. After obtaining complete histopathological diagnoses, the collected specimens were divided into two groups: 41 benign ovarian lesions and 38 ovarian epithelial cancers. Patients with the BRCA 1 mutation, those diagnosed with borderline ovarian tumors, and those with non-epithelial tumors were excluded from the study. Patients with epithelial non-serous types of ovarian cancer were excluded from the study.

The study protocol was approved on 22-02-2011 by the Pomeranian University of Medicine Ethical Committee number: KB-0012/58/11.

### 2.2. Immunohistochemical Analysis of Chosen Markers

Deparaffinized sections of ovaries (3 μM thick) were hydrated and heat epitope retrieval was performed using a microwave oven with retrieval solution buffer pH = 6 (DAKO, Dennmark). After cooling to room temperature (RT), the slides were incubated with 0.3% solution of H2O2, washed twice with PBS, and further incubated with 2.5% horse serum (Vector Laboratories, USA). After washing in PBS, the slides were incubated with primary antibodies: rabbit anti-human YAP (ProteinTech, Europe) and rabbit anti-human TEAD4 (Sigma-Aldrich, USA), rabbit anti-human ADH1A1 (ProteinTech, Europe), mouse anti-human CD71 (Invitrogen), and mouse anti-human H2A.X (Invitrogen) for 1h in RT. 

After washing in PBS, immunoreactions were visualized with ImmPRESS UNIVERSAL REAGENT and Vector NovaRED Substrate KIT FOR PEROXIDASE (VECTOR LABORATORIES, USA) according to the manufacturer’s protocol. As a negative control, the primary antibody was replaced with PBS on the specimen. Positive staining was defined by visual identification of a yellow/brown pigmentation in the light microscope. Images were collected with an Olympus IX81 inverted microscope (Olympus, Germany) with a color camera and with CellSens image processing software (Olympus, Germany).

### 2.3. Quantitative Real Time Polymerase Chain Reaction (qRT-PCR)

Quantitative analysis of the mRNA expression of *YAP1, SMAD2, SMAD3, CD71, H2A.X, TKTL1, ALDH1A1* and *TKT* genes was performed by two-step reverse transcription PCR. In our work, we examined the total protein expression. The total RNA was extracted from 50–100 mg tissue samples using a RNeasy Lipid Tissue Mini Kit (Qiagen). cDNA was prepared from 1 μg of total cellular RNA in 20 μL of reaction volume using a FirstStrand cDNA synthesis kit and oligo-dT primers (Fermentas). 

Quantitative assessment of the mRNA levels was performed by real-time RT-PCR using an ABI 7500 Fast instrument with Power SYBR Green PCR Master Mix reagent (Applied Biosystems). The conditions were as follows: 95 °C (15 s), 40 cycles at 95 °C (15 s), and 60 °C (1 min). According to melting point analysis, only one PCR product was amplified under these conditions. Each sample was analyzed in two technical replicates, and mean Ct values were used for further analysis. The relative quantity of a target, normalized to the endogenous controls GAPDH gene as internal calibrators, was calculated as the fold difference (2^dCt) and further processed using statistical analysis. The data are presented as the tumor tissue absolute expression.

### 2.4. Statistical Analysis

In the first stage of statistical analysis, the compliance of all of the obtained results with a Gaussian distribution was verified using the Shapiro–Wilk test. Most of the examined parameters had a non-normal distribution. Descriptive statistical parameters (the arithmetic mean, standard deviation, median, minimum, and maximum value) were calculated. The parameters calculated using semi-quantitative scales are represented by the median, the minimum (min), and the maximum (max) values. The results of the RT-PCR analysis were compared for the study group (A) and the control group (B). The Mann–Whitney U test was used in the case of unrelated variables (non-parametric test for unrelated variables for two groups).

Fisher’s exact test was conducted in order to compare two positive groups and the correlations between the expression of studied proteins and histological types or clinical-pathological features. The Wilcoxon rank-sum test was used for comparison between two immunohistochemical staining point groups.

## 3. Results

The mean age of the studied patient population was 54.2 years. The age range of the patients was 44–73 years. No age differences were found in the studied groups of patients. These characteristics are presented in Table 1.

After receiving the result of the histopathological examination, two groups of patients were created: the study group and the control group. The study group included 37 patients diagnosed with papillary serous carcinoma. The control group consisted of 41 patients.

Two subgroups were isolated in the group of patients without ovarian cancer: 21 patients with healthy ovaries and 20 patients with simple ovarian cysts. Almost all patients with ovarian cancer who qualified for the study were at the high stage of the clinical development of cancer. The exact characteristics of the patients are shown in Table 2.

### 3.1. Analysis of mRNA Expression for Individual Proteins

A significantly higher statistically significant level of mRNA expression was found in the ovarian cancer group for *YAP* and *TEAD4* (*p* = 0.003 and *p* = 0.006, respectively). There was no statistical significance in the expression of mRNA for *SMAD3*, and borderline statistical significance for *SMAD2* between the groups of ovarian cancer patients and other subgroups of patients with simple cysts and healthy ovarian tissue (*p* = 0.709 and *p* = 0.053, respectively). Moreover, a statistically significantly higher level of mRNA expression was found in the ovarian cancer group compared to the control group for the transferrin receptor (*CD71*), *H2A.X* and *ADH1A*. On the other hand, the differences in the mRNA expression for *TKT* and *TKTL1* were at a statistically significant level, with *p* = 0.003 and *p* = 0.016, respectively Table 3 and Table 4, Figure 1.

Immunohistochemical staining was performed on proteins that showed significant differences in expression during the qRT-PCR tests. We immunolocalized the *YAP*, *TEAD4, ADH1A1, CD71*, and *H2A.X* proteins in sections of ovarian tissue. *YAP* and TEAD4 showed strong nuclear and cytoplasmic staining in ovarian carcinoma and weak staining in non-carcinoma ovarian samples, *ADH1A1* showed strong staining in the cytoplasm of carcinoma sections and a weak positive reaction in the non-carcinoma section, *H2A.X* showed strong positive nuclear staining in carcinoma sections and moderate positive staining in non-carcinoma samples, and *CD71* showed moderate positive staining in carcinoma and non-carcinoma samples. The original magnification was 20× objective power, scale bar 50 μM (Figure 2, Table 5).

As shown in Table 6, we found no statistically significant differences in the expression of the tested proteins during the immunohistochemical test depending on age groups and grading (Figure 3). For two proteins, *YAP* and *TEAD4* higher expression was found in the case of higher clinical advancement of the tumor and in the presence of metastases.

### 3.2. Proteins as a Prognostic Factors

After analysis with the univariate Cox regression model, the strongest relationship with progression-free survival (PFS) and overall survival (OS) was confirmed for grading (respectively: *p* = 0.049; *p* = 0.008). We found a statistically significant relationship between longer PFS and protein: *YAP, TEAD4, SMAD2* and i concentration for the median (respectively: *p* = 0.031, *p* = 0.026, *p* = 0.048, *p* = 0.053, and *p* = 0.015). The relationship between longer overall survival and the *YAP, TEAD4, ADH.1, CD71* and *TKT* median was also respectively significant (*p* = 0.02, *p* = 0.017, *p* = 0.043, *p* = 0.024, and *p* = 0.028).

In the multivariate analysis, we found a relationship with both the progression-free survival and overall survival for only one protein. The *YAP* concentration was determined for the median or 95th percentile and correlated respectively with PFS (*p* = 0.027/*p* = 0.046) and OS (*p* = 0.041/0.039) as shown in Table 7.

## 4. Discussion

In 2020, there were approximately 313,000 new cases of ovarian cancer and 207,000 deaths due to this disease worldwide. The majority (65–75%) of cases were diagnosed with stage III or IV advanced neoplastic disease according to FIGO. Despite the original good response to chemotherapy treatment in 90% of patients, only about 15% of them did not experience a relapse after several years [4]. The most common histopathological type of ovarian cancer is the serous one. The endometrial, clear cell, or mucinous type is less common. The biology of ovarian cancer is highly heterogeneous and remains unexplained.

The main goal of any ongoing research is to attempt to find markers that would allow the early detection of tumors and more effective treatment, as well as longer patient survival. In this paper, we compared, on both the mRNA and protein level, a number of proteins from different effector pathways involved in apoptosis, proliferation, DNA repair, and immune processes; the aim was to identify those that could be used in the diagnosis or prognosis of ovarian cancer. 

One of the pathways that aroused our interest was the Hippo pathway [17], which is considered to be a pathway integrating extracellular and intracellular signals, resulting in the production and maintenance of the correct sizes of internal organs [18]. This pathway regulates organ size by controlling cell proliferation and apoptosis through the *YAP* protein, which is a transcription co-activator [19].

DNA damage or activation of the FAS death receptor leads to binding of the *YAP* protein during apoptosis [20]. In our study, the expression of *YAP* at both the mRNA and protein level was high in ovarian cancer tissue. The observed expression was low in benign cyst tissues. Similar reports were presented in studies by Yan Xia and Xio Wei, suggesting that the *YAP* protein may be directly associated with the development and progression of ovarian cancer [5,8,21]. 

The main elements of the Hippo pathway are kinases and adaptor proteins, which act by inhibiting the function of their nuclear effector—the YAP protein [6]. If the pathway ceases to be active, activated *YAP* migrates to the cell nucleus, where, along with the TEAD-family, transcription factors bind to gene promoter sites that are responsible for inhibiting apoptosis or promoting cell proliferation [22]. Attempts have been made to use the mechanisms of Hippo pathway action by blocking kinases in the regulation of the immune response in cancer immunotherapy [23].

Numerous literature data confirm the role of the components of the Hippo pathway, including both *YAP* and *TEAD* in tumorigenesis in various types of tissues: stomach, large intestine, breast, and esophagus [24,25,26]. In our study, we also confirmed the high expression of not only YAP but also TEAD at both the mRNA and protein level in ovarian cancer tissue. The clearly high expression in ovarian cancer tissue in both well-differentiated and poorly-differentiated cancers shows the possibility of using these proteins in an ovarian cancer diagnosis.

Another group of proteins with a prominent role in modern oncology is the SMAD-family proteins. These are responsible for transmitting the signal induced by TGF-beta to the cellular nucleus [27]. Changes to this pathway have been described in many diseases. The reduced expression of SMAD proteins has been observed in cardiovascular diseases, autoimmune diseases, Alzheimer′s disease, and osteoporosis [28]. Mutations of SMAD proteins have also been described in neoplastic diseases, mainly of epithelial origin from the gastrointestinal tract, lungs, and breasts [29]. 

The data on the role of SMAD proteins in the carcinogenesis of ovarian cancer appear to be inconclusive. Wang et al. noted that TGF-β1 inhibits ovarian tumor growth and improves sensitivity to chemotherapy by promoting the *BRCA1/SMAD3* signal pathway [30]. Kennedy et al. emphasized the possibility of excessive proliferation in ovarian cancer caused by abnormalities in the TGF-β and SMAD4 signaling pathway. In vitro studies on cell lines demonstrated that the increased expression of SMAD4 inhibits the migration and proliferation of ovarian cancer cells. The authors suggested that SMAD4 may inhibit the invasion and metastases of human ovarian cancer cells through a pathway that is also mediated by TGF-β [7]. 

In their studies on SKOV3 and CaOV3 ovarian cancer cell lineages, Dunfield et al. showed the key role of TGF β/SMAD disorders but described SMAD protein roles as uncertain and requiring further studies [2]. In our study, we found no difference in the mRNA expression for *SMAD3* and *SMAD4* between ovarian cancer tissue and simple cyst tissue. However, we must consider that it is difficult to make a comparison between the patient′s complex body environment, in which cells are present with a relatively “simple” environment with no additional signals, such as with an in vitro culture. Wakahara et al. reported the inhibition of proliferation by SMAD in SKOV3 cells in unstimulated culture [31].

### 4.1. Selected Factors Inducing DNA Damage

Shigeta et al. presented transferrin as a new biomolecule inducing DNA damage. In our study, we decided to investigate the expression of the transferrin receptor (CD71) in ovarian cancer patients and the expression of phosphorylated H2A.X protein during DNA damage [32].

Double-strand DNA breaks (DSB) are one of the most dangerous forms of DNA damage in the body, leading to either apoptosis or the formation of chromosomal aberrations. The main mediator of *H2A.X* phosphorylation occurring as a result of double-strand DNA breaks is ATM kinase [33,34].

ATM activation and *H2A.X* phosphorylation are related to *p53* protein function. The phosphorylation of *p53* in the Ser15 position causes its direct bonding to the sites of DNA fracture, likely facilitating the DNA repair processes [35]. Cells containing the normal *p53* protein usually have an elevated level of constitutive *H2A.X* phosphorylation compared to cells with a mutated form of this protein. This suggests that *p53* facilitates *H2A.X* phosphorylation, which mobilizes DNA repair systems against the formation of DSB, thus, preventing oxidative damage to DNA [36].

The current prevailing theory of neoplastic lesion formation concerns *p53* gene mutation. This mutation leads to pre-neoplastic lesions [37]. The presence of *p53* protein mutations and changes in the number of *H2A.X* gene copies were found in breast cancer; changes in the *H2A.X* promoter were associated with a higher risk of breast cancer and non-Hodgkin′s lymphoma [38]. Elevated levels of endogenous H2A.X, similarly to other proteins involved in DDR (DNA′s damage repair) were observed in people in pre-neoplastic conditions of the breast [39]. 

Changes of the DNA damage repair type occur in the entire population of patients, regardless of the presence or absence of germinal mutations. In our study, we found a significant difference in the expression of mRNA for *H2A.X*, whereas the differences were not so apparent in the immunohistochemical examination of proteins in ovarian tissue. The expression in the ovarian cancer tissue was moderate, while the expression was much weaker in the benign cyst tissue.

### 4.2. The Effect of Transferrin

Transferrin, through the TRF1 receptor, may cause the formation of double-strand DNA breaks; it also inhibits H2A.X phosphorylation, which stops the helix repair system [32]. Transferrin is one of the main glycoproteins in human serum and is undoubtedly essential for survival. Nevertheless, periodic, recurring exposure to higher levels of transferrin can promote cancer development, including poorly-differentiated ovarian cancer [32]. 

Our studies confirmed differences in the transferrin receptor mRNA levels between ovarian cancer tissue and benign cyst tissue. However, the differences in the qualitative expression in IHC were no longer as pronounced. In future studies, we intend to investigate the differences in the expression of the H2A.X protein and receptor protein for transferrin in the ovarian tissue of patients with a *BRCA1* mutation and healthy controls. Nearly all patients with a *BRCA1* mutation develop G3 grade ovarian cancer with poor histopathological differentiation, where differences in the expression of the examined proteins should be even more pronounced.

### 4.3. Effect of Alcohol

Frequent alcohol consumption can lead to numerous tumors. In particular, these include gastrointestinal tumors of the liver, the pancreas, and large intestine. In our study, we evaluated the expression of class 1 alcohol dehydrogenase (*ADH1A*) in ovarian cancer tissue and benign and healthy ovary tissue. We found high expression at both the mRNA and protein level in the ovarian cancer tissue, with little or no expression in the benign cyst tissue and healthy ovarian tissue. Similar reports were presented by Orywal et al. [40]. 

They showed a significant expression of mRNA for *ADH1A* in ovarian cancer tissue compared to healthy ovarian tissue and benign cysts [41]. However, they noted that the activity of the enzyme responsible for *ADH1A* degradation was the same in ovarian cancer tissue as in healthy tissue. These results suggest an increased capacity to produce *ADH1A* by ovarian cancer cells. Acetaldehyde interferes with DNA synthesis and repair by inhibiting methylguanine methyltransferase (MGMT) [42]. The higher expression of ADH1A was described in tissues in other types of neoplastic diseases: colorectal, uterine, and cervical cancer [43], while Jelski et al. described low ADH1A expression in breast cancer [11].

A high expression of ALDH aldehyde dehydrogenase was described in breast cancer, with the foregoing expression correlating with a poor prognosis and the total survival time [44]. Significantly increased expression of ADH1A in ovarian cancer tissue, with disproportionately unincreased levels of aldehyde dehydrogenase (degradation enzyme), may lead to the increased production of acetaldehyde and the support of carcinogenesis in ovarian tissue [45].

### 4.4. Transketolase Family

We selected two proteins from the transketolase family: TKT and TKTL2. These proteins are thiamine-dependent enzymes and are considered to be key enzymes in the non-oxidant part of the pentose phosphate pathway. This pathway provides over 85% of the ribose for nucleic acid synthesis, including in neoplastic cells. In our study, the differences in mRNA expression were statistically significant for *TKT* and *TKTL1*. 

Other publications on ovarian cancer confirmed the increased expression of these two proteins in ovarian cancer tissue compared to healthy tissue [46,47]. Studies conducted by Menghuang Zhao et al. not only showed the higher expression of *TKT* and *TKTL1* in ovarian cancer but also the high mRNA expression of the aforementioned transketolase genes, which is associated with a shorter PFS in patients with high clinical grade tumors [48]. A close correlation of TKTL1 expression with the overall survival (OS) was also demonstrated in all ovarian cancer patients.

To summarize our study, in which we traced several pathways involved in cell apoptosis and proliferation, the most promising appeared to be the YAP and TEAD4 complex, as well as alcohol dehydrogenase. The remaining proteins require further studies and clarification of their applications.

## 5. Conclusions

The YAP, TEAD4, and ADH1A proteins appear to be promising biomarkers in the diagnosis of ovarian cancer. The H2A.X, CD71, SMAD3 and 4, TKT, and TKTL1 proteins require further studies to confirm their suitability for clinical use in ovarian cancer patients. The YAP protein may be considered as a good prognostic factor in patients with ovarian cancer.

## Figures and Tables

**Figure 1 diagnostics-11-01026-f001:**
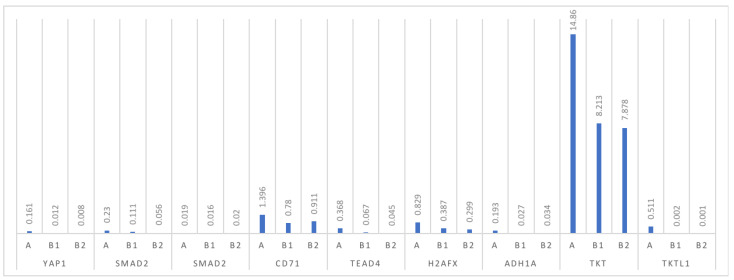
The expression distribution of the tested proteins.

**Figure 2 diagnostics-11-01026-f002:**
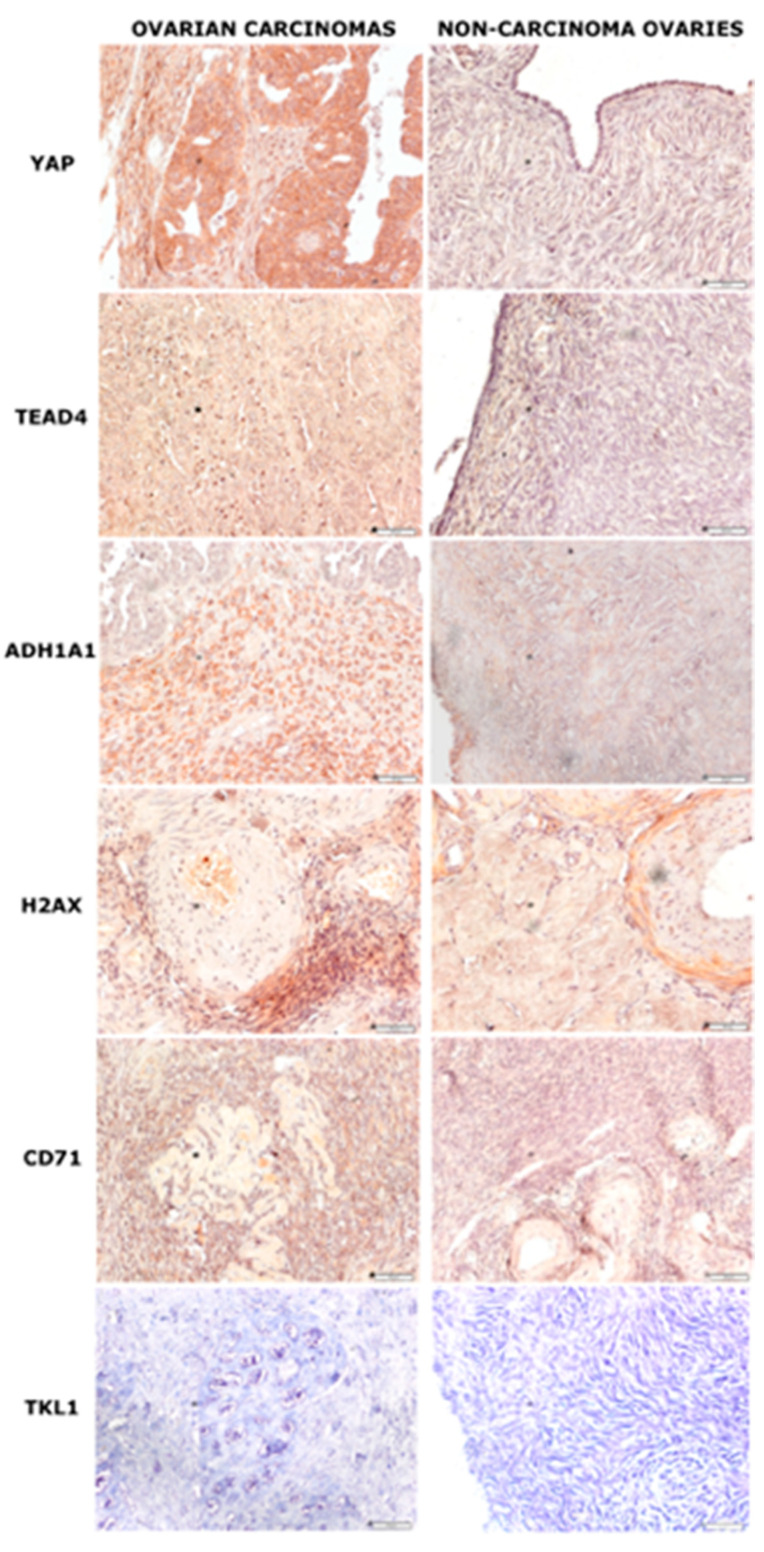
Immunohistochemical staining in ovarian carcinoma and non-carcinoma ovarian samples. Magnification: 20×, scale bar: 50 μM.

**Figure 3 diagnostics-11-01026-f003:**
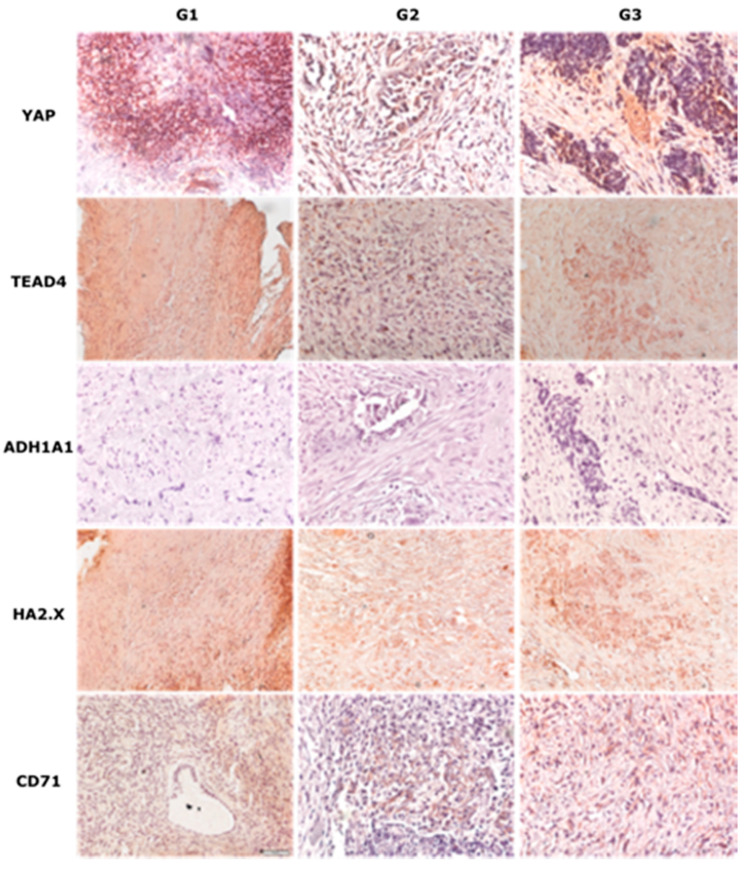
Immunohistochemical examination of YAP, TEAD4, ALDH1A1, HA2.X, and CD71 on specimens in different stages of disease (G1–G3). Magnification: 20×, scale bar 50 μM.

**Table 1 diagnostics-11-01026-t001:** The characteristics of each group and subgroup of patients.

Histopathology Type	*n Patient*	Age Mean (Years) (SD)	*p*
Papillary serous carcinoma	38	63.8 (1.6–2.9)	NS
Benign changes	41	50.3 (1.1–1.9)	
Simple cysts	21	44.9 (0.7–1.8)	NS
Normal ovary	20	58.2 (1.4–3.1)	

NS: not significant.

**Table 2 diagnostics-11-01026-t002:** Descriptive characteristics of patients with ovarian cancer.

	*n* Patient	Age Mean (Years) (SD)
Papillary serous adenocarcinoma	38	63.8 (2.1–3.0)
G1	6	66.6 (1.7–3.2)
G2	14	61.5 (2.8–4.1)
G3	18	62.3 (2.0–2.9)
FIGO I–II	2	60.2 (1.6–3.2)
FIGO III–IV	36	67.1 (2.4–3.4)

**Table 3 diagnostics-11-01026-t003:** Statistic parameters; group A: cancer and group B: non-cancer.

Variable	Statistics Parameters								
	*n Patient*	Mean	−95% CI	+95% CI	Median	Minimum	Maximum	Standard Deviation	*p*
YAP1 group A	41	0.164	0.054	0.272	0.048	0.001	1.561	0.345	0.003
YAP1 group B	40	0.011	0.005	0.016	0.005	0.000	0.091	0.0016	
SMAD2 group A	41	0.232	0.052	0.413	0.061	0.001	3.334	0.579	0.053
SMAD2 group B	40	0.059	0.001	0.116	0.019	0.000	1.000	0.171	
SMAD3 group A	41	0.018	0.011	0.026	0.010	0.001	0.351	0.092	NS
SMAD3 group B	40	0.081	–0.064	0.225	0.005	0.000	0.106	0.002	
CD71 group A	41	1.339	0.752	1.925	0.378	0.017	7.501	1.880	0.048
CD71 group B	40	0.887	0.403	1.371	0.167	0.006	7.412	1.533	
TEAD4 group A	41	0.371	0.200	0.542	0.088	0.011	2.351	0.548	0.006
TEAD4 group B	40	0.061	0.042	0.171	0.016	0.000	0.999	0.202	
H2AX group A	41	0.826	0.424	1.217	0.222	0.013	5.864	1.258	0.023
H2AXgroup B	40	0.511	0.195	0.829	0.110	0.017	5.671	1.002	
ADH1A group A	41	0.184	0.083	0.284	0.086	0.001	1.700	0.313	0.003
ADH1A group B	40	0.039	0.017	0.061	0.006	0.000	0.305	0.066	
TKT group A	41	14.861	9.241	20.497	7.564	1.108	162.113	18.066	0.003
TKT group B	40	7.933	–0.241	16.092	1.842	0.008	6.280	25.885	
TKTL1 group A	41	0.508	–0.053	1.071	0.001	0.000	6.681	1.411	0.016
TKTL1group B	40	0.002	–0.001	0.006	0.000	0.000	0.051	0.018	

NS: not significant. As many as 82 patients were qualified for immunohistochemical analysis.

**Table 4 diagnostics-11-01026-t004:** Statistic parameters; group B1: simple cyst and group B2: normal ovary.

Variable	Statistics Parameters								
	*n Patient*	Mean	−95% CI	+95% CI	Median	Minimum	Maximum	Standard Deviation	*p*
YAP1 group B1	21	0.012	0.021	0.040	0.009	0.001	1.002	0.021	NS
YAP1 group B2	20	0.008	0.024	0.031	0.006	0.000	0.893	0.009	
SMAD2 group B1	21	0.111	0.001	0.116	0.019	0.000	1.000	0.171	0.04
SMAD2 group B2	20	0.056	0.001	0.078	0.009	0.000	1.002	0.087	
SMAD3 group B1	21	0.016	0.012	0.033	0.010	0.000	0.311	0.078	NS
SMAD3 group B2	20	0.020	0.014	0.061	0.014	0.001	0.408	0.068	
CD71 group B1	21	0.780	0.356	1.206	0.423	0.006	7.112	1.432	NS
CD71 group B2	20	0.911	0.423	1.523	0.622	0.008	7.004	1.321	
TEAD4 group B1	21	0.067	0.042	0.171	0.016	0.000	0.999	0.202	NS
TEAD4 group B2	20	0.045	0.039	0.121	0.023	0.002	1.003	0.187	
H2AX group B1	21	0.387	0.109	0.912	0.110	0.012	5.076	0.976	NS
H2AX group B2	20	0.299	0.123	0.867	0.132	0.001	5.231	0.954	
ADH1A group B1	21	0.027	0.020	0.049	0.016	0.000	0.298	0.066	NS
ADH1A group B2	20	0.034	0.022	0.061	0.023	0.001	0.311	0.052	
TKT group B1	21	8.213	0.876	17.344	1.997	0.009	7.243	23.435	NS
TKT group B2	20	7.878	0.767	15.231	1,878	0.013	7.006	20.878	
TKTL1 group B1	21	0.002	0.001	0.005	0.000	0.000	0.051	0.018	NS
TKTL1 group B2	20	0.001	0.001	0.006	0.000	0.000	0.043	0.020	

NS: not significant.

**Table 5 diagnostics-11-01026-t005:** Comparison of the tested proteins immunohistochemical expression.

	Ovarian Cancer	Non Ovarian Cancer	*p*
**YAP**	POS + + +	POS -	<0.05
	NEG −	NEG +	
**TEAD4**	POS +++	POS −	<0.05
	NEG −x	NEG +	
**ADH1A1**	POS +++	POS +	<0.5
	NEG −	NEG +	
**H2AX**	POS ++	POS ++	NS
	NEG +	NEG +	
**CD71**	POS ++	POS +	NS
	NEG +	NEG +	
**TKL1**	POS +	POS +	NS
	NEG +	NEG −	

NS: not significant.

**Table 6 diagnostics-11-01026-t006:** Comparison of the tested proteins immunohistochemical expression depending on grading and staging.

Proteins	Age	*p*	Grade	*p*	FIGO	*p*	Metastasized	*p*
**YAP**	<55	NS	G1	NS	I,II	0.01	Yes	0.04
	>55		G3		III,IV		No	
**TEAD4**	<55	NS	G1	NS	I,II	0.01	Yes	0.02
	>55		G3		III,IV		No	
**ADH1A1**	<55	NS	G1	NS	I,II	NS	Yes	NS
	>55		G3		III,IV		No	
**H2AX**	<55	NS	G1	NS	I,II	NS	Yes	NS
	>55		G3		III,IV		No	
**CD71**	<55	NS	G1	NS	I,II	NS	Yes	NS
	>55		G3		III,IV		No	
**TKL1**	<55	NS	G1	NS	I,II	NS	Yes	NS
	>55		G3		III,IV		No	

NS: not significant.

**Table 7 diagnostics-11-01026-t007:** Univariant and multi variant analyses (Cox regression model).

	Univariate Analysis (Cox Regression Model)					
	PFS			OS		
	HR	95% CI	*p*-Value	HR	95% CI	*p*-Value
**Age**	1.019	0.78–1.22	0.061	1.21	0.99–1.26	0.037
**Grade 1 vs 3**	2.04	1.68–2.43	0.049	2.18	1.74–2.25	0.008
**YAP median**	1.43	1.21–1.62	0.031	1.40	1.33–1.48	0.02
**TEAD4 median**	1.28	1.11–1.47	0.026	1.09	0.97–1.30	0.017
**SMAD2 median**	1.06	1.02–1.21	0.048	1.16	1.04–1.27	NS
**SMAD3 median**	0.98	0.90–1.13	NS	0.89	0.72–0.99	NS
**H2A.X**	0.94	0.85–1.17	NS	0.97	0.76–1.08	NS
**ADH1.A**	1.23	1.07–1.32	NS	1.24	1.12–1.31	0.043
**CD71**	1.11	0.82–1.30	0.053	1.37	1.18–145	0.024
**TKT**	1.08	0.88–1.21	0.015	1.15	0.99–1.17	0.028
**TKTL1**	1.37	1.14–1.44	NS	1.26	1.18–1.38	NS
	**Multivariate Analysis (Cox Regression Model)**					
	**PFS**			**OS**		
	**HR**	**95% CI**	***p*-Value**	**HR**	**95% CI**	***p*-Value**
**YAP median**	1.22	1.06–1.31	0.027	1.19	0.98–1.24	0.041
**YAP 95 percentile**	1.49	1.36–1.59	0.046	1.31	1.26–1.39	0.039
**TEAD4 median**	1.10	0.95–1.27	0.007	1.17	1.08–1.32	NS
**TEAD4 95 percentile**	1.34	1.26–1.43	NS	1.02	0.94–1.22	NS
**CD71 median**	1.21	1.11–1.36	0.033	1.37	1.24–1.43	NS
**CD71 95 percentile**	1.08	0.96–1.21	NS	1.19	1.12–1.34	0.018
**TKT median**	0.99	0.90–1.17	NS	1.07	0.97–1.22	NS
**TKT 95 percentile**	1.12	1.01–1.28	0.004	1.28	1.13–1.32	NS

NS: not significant.

## Data Availability

The data sets used during current study are available from the corresponding author on reasonable request.

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
