# Peer review of "Importance of Potential New Biomarkers in Patient with Serouse Ovarian Cancer"

_diagnostics, 2021, doi:10.3390/diagnostics11061026_

Round 1

Reviewer 1 Report

1. The study group includes 37 patients with serous ovarian cancer, 2 patients with endometrial ovarian cancer and 1 patient with clear-cell ovarian cancer. I would recommend that the authors exclude all patients except those with serous ovarian cancer and clarify the title of the article. From a statistical point of view, the inclusion of such small groups is not legitimate. 2. There is no data on the presence / absence of statistical differences in age between cancerous and non-cancerous subgroups. 3. Why did you not assess the relationship between the study parameters and survival rates? If such data are available, it would be appropriate to show them. 

Author Response

Reviewer 1

Thank you very much for your valuable tips. Ovarian cancer is a cancer where we cannot talk about prevention, even in high-risk groups. Therefore, the search for new biomarkers that would improve diagnostics seems to be crucial.

The study group includes 37 patients with serous ovarian cancer, 2 patients with endometrial ovarian cancer and 1 patient with clear-cell ovarian cancer. I would recommend that the authors exclude all patients except those with serous ovarian cancer and clarify the title of the article. From a statistical point of view, the inclusion of such small groups is not legitimate

1.Of course, we agree with the reviewer.

We removed the cases of patients with other histopathological types than serous ovarian cancer. We corrected the title of the work and the statistical analysis was performed again.

  1. There is no data on the presence / absence of statistical differences in age between cancerous and non-cancerous subgroups.

We supplemented the tables and the text with data showing the lack of statistical significance between these groups and subgroups.

  1. Why did you not assess the relationship between the study parameters and survival rates? If such data are available, it would be appropriate to show them.

We have follow up of patients. As suggested by the reviewer, we added univariant and multivariant analysis in the results section. We changed the work goals and added an additional application.

Reviewer 2 Report

Thank you very much for the opportunity to review this interesting paper.
Due to the high mortality rate of ovarian cancer, it is very important
to detect it quickly. Currently, the research on tumor markers is insufficient
and most often non-specific. That is why, in my opinion, every small step
towards improving the diagnosis of ovarian cancer is important for the
development of medicine.
As we all know, early detection is of the greatest
importance in ovarian cancer. Currently, ovarian cancer is most often detected in stage 3 and 4, which means that the treatment is ineffective. Of course, I have a few comments about the work, which I hope will improve its quality even more:
1. References:
a. N
on-English-language literature should not be used in an international journal, so please replace these journals with those available at WoS or Pubmed.

b. Due to the innovative design, the literature should be limited to 10 better 5 years. Cut-off date 2011 better 2016.

2. 

line 39-41 Epithelial ovarian cancers (EOCs) constitute the majority of malignant ovarian cancers in adult women. Based on 5 years of experience, we can conclude that they demonstrate the worst prognosis of all gynecological cancers. Conventional therapies usually include surgery and chemotherapy (platinum and taxol based, as well as radiotherapy).

-Only 5 years of experience is not enough :) and re-edit this sentence.

Line 43-49 Please show the source.

Line 112-113  Perhaps the histopathological type or grade should be given here? (For the authors' decision  table 1 and table 2).

Line 217-221 These are not data from 2017 - see GLOBOCAN and re-edit

Line 254-260 Is it separate the woman with cardiovascular disease, osteoporosis from study group if it include for SMAD family?

Line 312-313 353-356 Maybe better in conclusion - author`s decision

Line 334-338 Is it only in genetically determined cancers or all?

Author Response

Thank you very much for your valuable tips that will definitely improve the quality of our work

  1. References:
    a. Non-English-language literature should not be used in an international journal, so please replace these journals with those available at WoS or Pubmed.
  2. Due to the innovative design, the literature should be limited to 10 better 5 years. Cut-off date 2011 better 2016.

Of course, we agree with the reviewer, the Polish items have been removed and the older items in the journal with newer items have been added or replaced

  1.  

line 39-41 Epithelial ovarian cancers (EOCs) constitute the majority of malignant ovarian cancers in adult women. Based on 5 years of experience, we can conclude that they demonstrate the worst prognosis of all gynecological cancers. Conventional therapies usually include surgery and chemotherapy (platinum and taxol based, as well as radiotherapy).

-Only 5 years of experience is not enough :) and re-edit this sentence.

- the sentences have been redrafted as suggested by the reviewer

Line 43-49 Please show the source.

- citation source added

Line 112-113  Perhaps the histopathological type or grade should be given here? (For the authors' decision  table 1 and table 2).

- according to the suggestions of the first reviewer, other histopathological types of the tumor were removed from the publication

Line 217-221 These are not data from 2017 - see GLOBOCAN and re-edit

-Citation has been changed to the GLOBOCAN data from 2020

Line 254-260 Is it separate the woman with cardiovascular disease, osteoporosis from study group if it includes for SMAD family?

The patients enrolled in the study had no cardiac problems, apart from hypertension, but without myocardial remodeling. There was one patient with osteopenia, but this should not interfere with the results

Line 312-313 353-356 Maybe better in conclusion - author`s decision

- this section was not transferred to the conclusions

Line 334-338 Is it only in genetically determined cancers or all?

It was added in the text that this applies to the entire patient population

Round 2

Reviewer 1 Report

I have no more comments on the article.